# Automated reporting of cervical biopsies using artificial intelligence

**Mahnaz Mohammadi**[1]*, **Christina Fell**[1], **David Morrison**[1], **Sheeba Syed**[2],
**Prakash Konanahalli**[2], **Sarah Bell**[2], **Gareth Bryson**[2], **Ognjen Arandjelović**[1], **David J. Harrison**[3,4], **David Harris-Birtill**[1]

1 School of Computer Science, University of St Andrews, St Andrews, United Kingdom, 2 Department of Pathology, Queen Elizabeth University Hospital, Glasgow, United Kingdom, 3 School of Medicine, University of St Andrews, United Kingdom, 4 Pathology, Division of Laboratory Medicine, Royal Infirmary of Edinburgh, United Kingdom

* mm459@st-andrews.ac.uk

**Data Availability Statement:** The data is available at https://doi.org/10.5281/zenodo.10276838.

**Funding:** For all authors this work is supported by the Industrial Centre for AI Research in digital Diagnostics (iCAIRD) which is funded by Innovate

## Abstract

When detected at an early stage, the 5-year survival rate for people with invasive cervical cancer is 92%. Being aware of signs and symptoms of cervical cancer and early detection greatly improve the chances of successful treatment. We have developed an Artificial Intelligence (AI) algorithm, trained and evaluated on cervical biopsies for automated reporting of digital diagnostics. The aim is to increase overall efficiency of pathological diagnosis and to have the performance tuned to high sensitivity for malignant cases. Having a tool for triage/identifying cancer and high grade lesions may potentially reduce reporting time by identifying areas of interest in a slide for the pathologist and therefore improving efficiency. We trained and validated our algorithm on 1738 cervical WSIs with one WSI per patient. On the independent test set of 811 WSIs, we achieved 93.4% malignant sensitivity for classifying slides. Recognising a WSI, with our algorithm, takes approximately 1.5 minutes on the NVIDIA Tesla V100 GPU. Whole slide images of different formats (TIFF, iSyntax, and CZI) can be processed using this code, and it is easily extendable to other formats.

## Author summary

The majority of biopsies received by pathologists for reporting, do not contain invasive cancer. This yields opportunities for Artificial Intelligence (AI) development in identifying cancerous and high grade lesions in a slide and reduce the necessity of pathologist having to review the whole slide and negative biopsies. We have developed an Artificial Intelligence (AI) algorithm, trained and evaluated on cervical biopsies for automated reporting of digital diagnostics with the aim to increase overall efficiency of pathological diagnosis and to have the performance tuned to high sensitivity for malignant cases. This can potentially reduce reporting time and improve overall efficiency of pathological diagnosis.

UK on behalf of UK Research and Innovation (UKRI) [project number: 104690], and in part by Chief Scientist Office, Scotland. DJH, DHB, OA and GB received funding from UKRI (funder project reference: TS/S013121/1). MM, DM and CF received salaries from UKRI for this project. The funders had no role in study design, data collection and analysis, decision to publish, or preparation of the manuscript.

**Competing interests:** The authors have declared that no competing interests exist.

# 1 Introduction

Cervical cancer is the $4^{th}$ most common cancer worldwide and $14^{th}$ most common cancer in females in the UK [1], when detected at an early stage, the 5-year survival rate for people with invasive cervical cancer is 92% [2]. The most common types of cervical cancers are squamous cell carcinoma and adenocarcinoma. The majority of cases are associated with high risk human papillomavirus (HPV) infection. Most cases of cervical cancer are preceded by pre-invasive, dysplastic lesions (cervical intraepithelial neoplasia, CIN and cervical glandular intraepithelial neoplasia, CGIN) and CIN can be graded to determine the risk of progression to cancer and the need for further treatment. These abnormal cells can be detected by examining the lining of the cervix (cervical smear) and have formed the basis of cervical screening programmes throughout the world, allowing early detection, treatment, and prevention of invasive cancer. Following an abnormal smear result, the cervix is examined at colposcopy and tissue biopsies taken to identify treatable pre-invasive lesions and also to identify invasive carcinoma. The majority of biopsies received by pathologists for reporting do not contain invasive cancer which yields opportunities for Artificial Intelligence (AI) development in cancer detection to reduce the necessity of pathologist having to review the negative biopsies.

In modern clinical practice, digital pathology and its integration with AI has enabled true utilisation and integration of knowledge that is beyond human limits and boundaries [3]. In recent years, clinicians hope to take advantage of advances in digital imaging and machine learning (ML) to improve medical image analysis. ML algorithms have been of great help in many medical applications and can be used for early detection of cancerous regions [4–6]. However, manually extracting features (handcrafted), need expert domain knowledge and the procedure is laborious and time-consuming. The high-level feature representation of deep convolutional neural networks has proven to be superior to handcrafted low-level and mid-level features. The main advantage of the deep learning is that it can automatically learn data-driven (or task-specific), highly representative and hierarchical features, and performs feature extraction and classification on a network, which is trained in an end-to-end manner.

A study is undertaken to compare the accuracy of medical image classification among three types of machine learning models including Support Vector Machine (SVM), Artificial Neural Network (ANN), and Convolutional Neural Network (CNN) [7]. To investigate changes in accuracy related to image quality, a single dataset using two different file formats of DICOM (Digital Imaging and Communications in Medicine) and JPEG (Joint Photographic Experts Group) was constructed. CNN classification was accurate for both datasets even though the JPEG format contains less colour information and data capacity than the DICOM format, whereas SVM and ANN accuracy decreased with the loss of data from DICOM to JPEG formats.

Deep learning algorithms are applied to medical image analysis and used for classification and segmentation of medical images, including CT/MRI tomography, ultrasound, and digital pathology [8].

Recently, deep learning has become the mainstream methodological choice for analysing and interpreting histology images. Different machine learning and deep learning strategies such as supervised, weakly supervised, unsupervised, transfer learning and various other sub-variants of these methods [9] are applied to digital H&E-stained pathology images for colour normalization, nuclei/tissue segmentation, and cancer diagnosis and prognosis [10]. The experimental results of these studies demonstrates that deep learning is a promising tool to assist clinicians in the clinical management of human cancers.

Clinical-level-aided diagnosis system for cervical cancer screening based on deep learning are being investigated for efficient and high-accuracy predictions. Some studies built slide-

level classification systems by multi-stage designs. For example, Cheng et al. [11] designed a robust and progressive WSI analysis method for cervical cancer screening. In the first stage, the authors developed a progressive lesion cell recognition method combining low- and high-resolution WSIs. Then, a RNN-based WSI classification model was built for WSI-level predictions in the second stage.

A localized, fusion-based, hybrid imaging and deep learning approach [12] is introduced to classify squamous epithelium into Normal, CIN1, CIN2, and CIN3 grades of cervical intraepithelial neoplasia (CIN). The approach partitioned the epithelium into 10 segments and each segment into 3 parts (top, middle, bottom) and uses a CNN to classify the top, middle and bottom parts. The results are then fused to classify the segment and the whole epithelium.

Transfer learning using deep pre-trained convolutional neural networks is increasingly used to solve numerous problems in the medical field [13]. In this study, we use pretrained GoogleNet convolutional neural network for automated reporting of digital diagnostics within the pathology AI stream of Industrial Centre for Artificial Intelligence Research in Digital Diagnostics (iCAIRD). The aim is to increase overall efficiency of pathological diagnosis and to have the performance tuned to high sensitivity for cervical malignant cases. The algorithm is trained on 1738 cervical WSIs and evaluated on 801 WSIs in the test set and achieved 93.4% malignant sensitivity for WSI diagnosis. Following appropriate clinical validation and regulatory clearance, the developed algorithms could be integrated into clinical workflow. Having a tool for triage/identifying cancer and high grade lesions may potentially reduce reporting time by identifying areas of interest in a slide for the pathologist and therefore improving efficiency.

## 2 Materials and methods

In this section, we cover the data preparation and annotation process, data splitting and the overall structure of the classification algorithm and its steps in detail.

### 2.1 Data preparation and annotation process

The cervical tissue blocks for this study originate from Glasgow Royal Infirmary (NG), Southern General Hospital (SG), Royal Alexandria Hospital (RAH) and Queen Elizabeth University Hospital (QEUH) (all in Glasgow, Scotland) each with independent tissue handling including fixation and tissue processing. The number of tissue blocks obtained from each of the above sites were: 829 from QEUH, 729 from NG, 647 from SG and 334 from RAH.

New tissue sections were cut from the tissue blocks at one of two different thicknesses (3 microns or 4 microns) and then stained with one of four different H&E protocols (routine H&E, muscle biopsy protocol, neuro protocol and paeds protocol). Together, these combinations gave eight different labs maximising WSI variance and thereby decrease the likelihood of overfit to any one lab (combination of tissue processing, cutting and staining protocol).

All slides were then scanned at QEUH and saved as WSI. The WSI are hundreds of thousands of pixels in height and width at the highest magnification and are too large to read into memory. Dedicated WSI formats allow access to either small parts of the image at the highest magnification or the whole image at lower magnifications. For this study, the slides were scanned using a Phillips Ultra Fast Scanner (UFS) with resolution equivalent to 40x or more specifically 0.25 microns/pixel, and stored in the isyntax file format. The most detailed view in the WSI is level 0, or 40x magnification where the length of a side of 1 pixel in the image is 0.25μm. Higher levels represent lower magnifications in a pyramid where each level is a power of 2 smaller than the previous. For example at level 5, one pixel represents a square patch at level 0 with a length of $2^5 = 32$ pixels per side, or an area of $32 \times 32 = 1024$ pixels in total.

The annotation process stratified slides into four main diagnostic categories which include malignant, high grade, low-grade and Normal/inflammation. Each diagnostic category has sub-categories. The categories and their sub-categories are defined as follows:

1. Malignant: Squamous cell cervical cancer (SCC) and adenocarcinoma (AC) are the most common types of cervical cancer. Both of these are capable of local spread and metastasis. Cervical glandular intraepithelial Neoplasia (CGIN) is an uncommon pre-invasive dysplastic lesion of glandular cells which can develop into AC. There is histological overlap with some well differentiated AC and this lesion tends to be treated more aggressively.

2. High Grade: Cervical intraepithelial neoplasia (CIN) is graded to determine risk of development of cancer and to guide further management. Most countries have now moved to a two tier classification for CIN (high grade and low grade). In the UK, pathologists still often refer to the old three tier classification (CIN1/2/3). For the purposes of this algorithm we classified 'high grade' lesions as those with morphological features of CIN 2 or 3.

3. Low Grade: A slide is labelled as low grade if it contains slightly abnormal cells on the surface of the cervix (CIN 1) or low-grade changes that are usually caused by an HPV infection (HPV). CIN 1 and HPV are not cancer and usually go away on their own without treatment, but sometimes they can become cancer and spread into nearby tissue.

4. Normal/inflammation: Cervicitis is an inflammation of the cervix. Cervicitis is common and may be caused by a number of factors, including infections, chemical or physical irritation, and allergies. Normal tissue and cervicitis fall within this category.

All slides from benign and malignant cervical biopsies were digitally scanned on a Philips UFS Scanner as iSyntax Whole Slide Images (WSIs). WSIs were exported from the Philips information management system and converted to OME-Tiffs using (Glencoe Software) to make them compatible with QuPath [14] (Version v0.2.3) for annotation. The annotation procedure involved defining the main slide category, then manually annotating any additional subcategories that could have been available on the WSI. The annotation vector files were aligned to the original iSyntax images for analysis.

Each slide was randomly assigned to one of four participating Consultant Pathologists for annotation. Each of the participating pathologists had a sub-specialist interest in Gynaecological Pathology, and participated in the UK National Gynaecological Pathology External Quality Assurance Scheme. Primary annotation was performed either by one of the four pathologists, or by a biomedical scientist, who were specifically trained for this project. All annotations done by a biomedical scientist were signed off by one of the study pathologists.

## 2.2 Data splitting to train, validation and test sets

We received a total of 2539 whole slide images (WSIs), with only one slide per patient, in iSyntax format, an Excel file containing metadata, categories, and subcategories and an annotation file per WSI in JSON format at the end of the annotation process. These WSIs were split into training, validation and test sets as shown in Table 1.

The split percentages were calculated based on the case labels associated with the samples recorded in the system and the numbers per each set were agreed on by all the team members and the pathologists. All slides from two of the labs and 10% randomly selected slides from the other six labs were set aside as test set and never used in the training and validation process. The remaining 90% of the slides, from the 6 other labs were used as training and validation sets. Two third of these slides were randomly selected and used as validation set. To retain the the same proportion of classes in the train and test sets that are found in the entire original

**Table 1. Distribution of samples in training, validation, and test sets for iCAIRD cervical dataset.**

| Category | SubCategory | Count | Training | Validation | Test |
|---|---|---|---|---|---|
| Malignant | - Squamous carcinoma | 268 | 127 | 60 | 81 |
| | - Adenocarcinoma | 107 | 243 | 23 | 38 |
| | - CGIN | 92 | 41 | 19 | 32 |
| | - Other* | 59 | 29 | 15 | 15 |
| High Grade | - CIN 2 | 320 | 141 | 71 | 108 |
| | - CIN 3 | 321 | 146 | 75 | 100 |
| Low Grade | - HPV | 420 | 197 | 96 | 127 |
| | - CIN 1 | 362 | 169 | 84 | 109 |
| Normal/inflammation | - Normal/inflammation | 590 | 268 | 191 | 131 |
| Total | | 2539 | 1164 | 574 | 801 |

* Other subcategory in malignant are biopsies with a malignant diagnosis that don't fall under adenocarcinoma or squamous carcinoma. Examples are involvement of the cervix by endometrial tumours or metastases spread from tumours in other parts of the body or other types of malignant tumour that are not carcinoma (e.g. sarcoma).

dataset, dataset was split using a stratified fashion balanced over categories, subcategories and staining labs for training, validation and test sets. During the annotation process these labels were doubled checked and in approximately 5% of the cases the final label associated with the scanned slide was different. This could be because the new slice taken from the sample did not show the same pathology as the original or that the original label was incorrectly recorded. The corrected labels post annotation were the labels that were used for training and testing. This means the final numbers of slides of each type in Table 1 may not match the original percentages described above.

We are seeking to make the data fully and publicly available and we are navigating the necessary ethical approvals for releasing the data. Currently all the data for this study will be available on request for non-commercial and academic purposes from the director of iCAIRD (david.harrison@st-andrews.ac.uk).

## 2.3 Overall structure of the classification algorithm

Fig 1 shows the overall structure of the pipeline for cervical WSI diagnosis. This pipeline consists of different components, each described in details in following subsections.

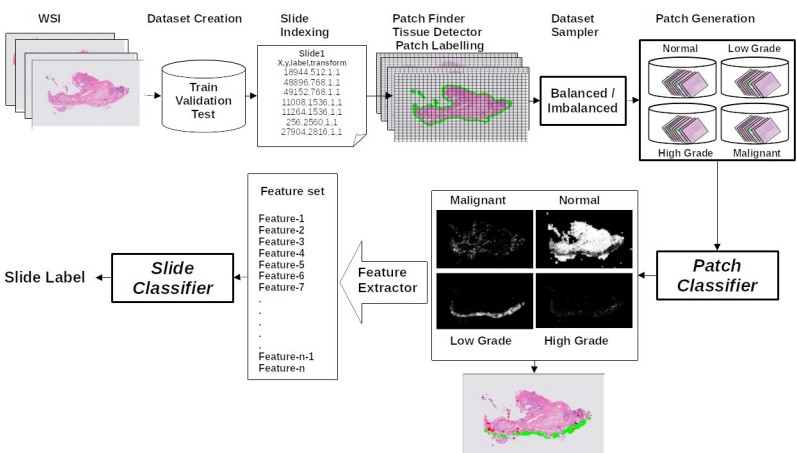

**Fig 1. Architecture of the pipeline for cervical whole slide image diagnosis.**

### 2.3.1 Dataset creation and patch generation.

**Dataset creation**. Slides and their respective annotation files are loaded. The dataset builder creates balanced training, validation, and test sets according to the dataset splitting rules described in section 2.2.

**Slide indexing**. This pipeline step generates index files for the training and validation datasets created in the previous step, making the subsequent pipeline processes easier. The paths to the slides, patch size, and the patch level at which patches are to be extracted are all contained in dataframes called index files.

**Patch finder**. WSIs are too large to be loaded to memory for further processing. Hence, using the paths in the index files, a low resolution version of each WSI (thumbnail) is generated to facilitate loading the entire image at once to the memory. To generate the downsampled version of the WSI, we need to define at which level of magnification ($mag\_level$) we want to read the WSI. When $mag\_level = 0$, we are reading the WSI at the original level without any downsampling. The width and height of the WSI are divided by $2^{(mag\_level)}$ for the thumbnail. Non-overlapping patches are then found on the thumbnail. For example to extract patches of size ($256 \times 256$) pixels from the original WSI at highest resolution, if the thumbnail is generated at level 5, the grid patch finder starts from the top left corner of the thumbnail and find $x$, $y$ coordinates of ($8 \times 8$) pixels regions from the thumbnail which are equivalent to ($256 \times 256$) pixel regions in the original slide. Each pixel within the thumbnail corresponds to a ($32 \times 32$) pixel region in the original slide. These coordinates (top left position of the patches found on the thumbnails) are then multiplied by $2^{(mag\_level)}$ to get their corresponding coordinates on the original WSI.

**Tissue detector**. Most of the slide is background (non-tissue) and does not contain useful information. To save on the computational cost, we remove background patches by applying a tissue detector to the patches found in the previous step to segment the background from the tissue. We follow a conservative method and label a patch as tissue, even if only one pixel in it is tissue. Patches are then labelled as tissue or non-tissue to make it convenient to remove the non-tissue patches from the dataset.

**Patch labelling**. Using the pixel level annotations provided by pathologists, each tissue patch is then labelled with one of the main categories (malignant, high grade, low grade and normal/inflammation). A patch with even one pixel from a most severe category (i.e. malignant is the most severe category, followed by high grade, low grade and normal) will be labelled as that most severe category (i.e. if only one pixel of a tissue patch is malignant and rest are high-grade or any other category, that patch is labelled as malignant which is the most severe category). A dataframe per slide containing the coordinates of the tissue patches and their categories is the output of this step of the pipeline.

**Dataset sampling**. The dataframes from the previous step are combined to form a single dataframe for the whole dataset. This makes it easier to know how many patches from each category exist in the whole dataset. We can sample a balanced subset, or we can declare the maximum number of patches to sample from each category. If the number of patches in a category is less than the maximum number specified, then all the patches from that category will be sampled. For the categories with more patches than the specified maximum number, patches are sampled randomly.

**Patch generation**. Sampled patches for each category are physically generated and placed in the folders with the same name as the categories defined for training and validation slides. Physically generating patches at this step removes the need to load the WSI each time a patch from that slide is accessed during training. It also saves on disk space by removing the background patches and reduces the computational cost of processing patches with no useful information further.

**2.3.2 Patch level training and inferencing.** After creating a dataset, we train the model on the training and validate it on validation patch sets. The type of the model used, and its parameters are described as follows:

**Patch classifier**. The patch classifier used is the pretrained standard *GoogLeNet* from Torchvision library.

**Data transformation**. We augment the patch sets by applying few transformations from *torchvision.transforms* module. We rotate the patches randomly by 0˚, 90˚, 180˚, 270˚ angles. We change the brightness, contrast, saturation, and hue of the patches by ColorJitter(*brightness* = 0.25, *contrast* = 0.75, *saturation* = 0.25, *hue* = 0.04). Finally, we convert the patches to PyTorch tensors and normalise the image using Normalize((0.5, 0.5, 0.5), (0.5, 0.5, 0.5)), based on mean and standard deviation in each colour channels in the range [−1, 1].

**Loss function**. If the dataset used for training is balanced over the categories, *CrossEntropyLoss* is used as loss function, otherwise *FocalLoss* [15] is used to address class imbalance by applying a modulating term to the cross entropy loss in order to focus learning on hard misclassified examples.

**Optimiser and scheduler**. *SGD optimiser* with *momentum* = 0.9, *learningrate* = 0.001 and *weightdecay* = 0.0005 is used to update the model parameters based on the computed gradients. A learning rate scheduler is used to adjust the learning rate during training. After every 2 epochs, the learning rate is reduced by a factor of learning rate (*gamma* = 0.5).

**Parallelism**. *DistributedDataParallel* (DDP) from PyTorch Lightning implements data parallelism at the module level and can be used for training on multiple GPUs. Since we are running our training on NVIDIA DGX-1, we use this plugin to speed up training by using multiple GPUs.

**Early stopping**. The model is trained for maximum of 20 epochs. Early stopping is used to avoid overfitting while training by monitoring the validation accuracy. If there is no improvement in the validation accuracy after 10 iterations, the training procedure is terminated and the best model with higher validation accuracy is saved.

**Inference on training and validation datasets and computing patch level results**. The trained patch classifier model is used to make predictions on training and validation sets. This is accomplished by evaluation of the model on all the tissue patches in each slide despite they are used for training or not. Predictions are probabilities per category for each patch on the slide. A binary heatmap is generated per category per slide using the patch probabilities. Higher probabilities are shown as brighter pixels in a heatmap. The computed probabilities are used to compute the final prediction at patch level for each slide and to create the patch level confusion matrices for training and validation datasets.

Figs 2, 3, 4, 5 and 6 are examples of ground truth labels provided by the pathologists, the generated patch level heatmaps for each category (normal, low grade, high grade, and malignant), and a prediction heatmap (i.e. the patches predictions based on a specified threshold) for different categories of slides. In all these examples, we are showing *probabilities* $\geq$ 0.5 on the generated heatmaps. We discuss these heatmaps in detail in section 4.

**2.3.3 Slide level classification.** There are different machine learning or deep learning classifiers that can be used as a slide level classifier [16]. We applied *Random Forest* and *XGBoost* [17] classifiers separately to the features to get the final slide level diagnosis. We extract the features from the heatmaps using *skimage.measure.regionprops*. The features extracted from the heatmaps generated at patch level are used for training to form the final slide level predictions for each slide. The features extracted are a refined combination of features stated in [18–20] which are extracted for heatmaps generated per class.

**Extracting features for Random Forest (RF) slide level classifier**. To generate the features for RF classifier, the heatmaps generated for the categories except for normal category (i.e.

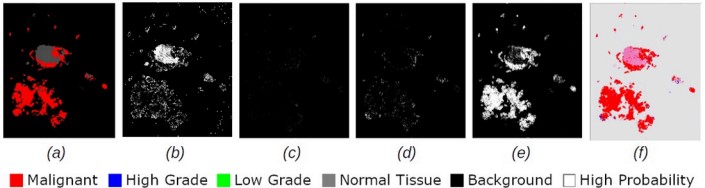

**Fig 2. Patch level heatmaps for a malignant slide.** (a):Truth Label (b): Normal (c):Low Grade (d): High Grade (e): Malignant (f): Prediction.

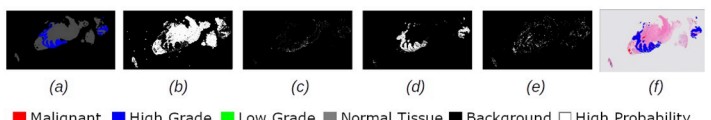

**Fig 3. Patch level heatmaps for a high grade slide.** (a):Truth Label (b): Normal (c):Low Grade (d): High Grade (e): Malignant (f): Prediction.

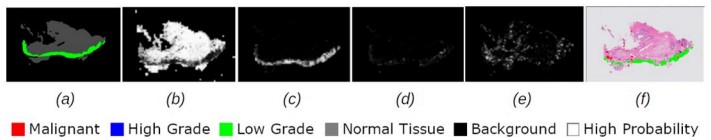

**Fig 4. Patch level heatmaps for a low grade slide.** (a):Truth Label (b): Normal (c):Low Grade (d): High Grade (e): Malignant (f): Prediction.

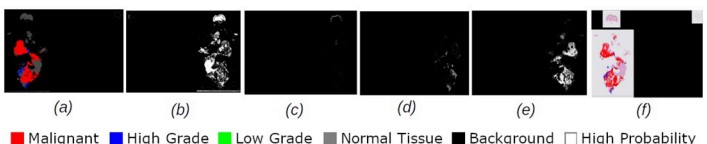

**Fig 5. Patch level heatmaps for a multilabel malignant slide.** (a):Truth Label (b): Normal (c):Low Grade (d): High Grade (e): Malignant (f): Prediction.

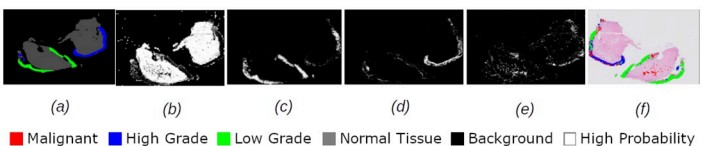

**Fig 6. Patch level heatmaps for a multilabel high grade slide.** (a):Truth Label (b): Normal (c):Low Grade (d): High Grade (e): Malignant (f): Prediction.

malignant, high grade, low grade) per slide, are thresholded at a series of different levels (0.5, 0.6, 0.7, 0.8, 0.9) to produce a set of binary images. For each binary image, the following global features are then computed:

- area_ratio: the ratio of number of pixels over the given probability to the tissue area.

- prob_area: the sum of probability of all the pixels over the threshold divided by the tissue area.

This results in 2 features at each threshold level for each of the three heatmaps. Hence, a total of 30 global features from three heatmaps at 5 different threshold levels are extracted per slide. Connected-component analysis is then applied to split the heatmaps into regions. Connect-component analysis was implemented using the *ConnectedComponents* from scikit-image. For the two largest regions, based on number of pixels, the following 10 regional features are extracted using the *regionprops* function from scikit-image.

- area of the region

- the eccentricity of an ellipse that has the same second moments as the region

- the ratio of the area to the area of the bounding box

- the bounding box area

- the major axis length of the ellipse with the same second moments of area as the region

- the max intensity within the region

- the mean intensity within the region

- the min intensity within the region

- the aspect ratio of the bounding box

- the ratio of the area of the region to the area of the convex area

This resulted in a total of 60 features extracted from the 2 largest regions on each of malignant, high grade and low grade categories of heatmaps. Combining the global and regional features together gives a 90-dimensional feature vector per slide, to be used in training for the slide level classifier.

**Extracting and refining features for XGBoost slide level classifier**. In order to refine the features set and reduce the dimension, we conducted experiments to identify correlated features. These experiments led to removal or replacement of some global and regional features with some more useful features.

To generate global features for XGBoost classifier, the heatmaps generated for the two categories (i.e. malignant, high grade) per slide are thresholded at 0.5, 0.7, 0.9 levels to produce a set of binary images. For each binary image, only the "area_ratio" (i.e. the ratio of number of pixels over the given probability to the tissue area) is computed as a global feature. This resulted in a total of 6 global features per slide. Connected-component analysis is then used to detect 7 largest malignant, 5 largest high grade, 3 largest low grade, and 2 largest normal regions from the heatmaps of each category. The regional features were extracted from the detected regions. Since the trials revealed that some features did not significantly contribute to dividing the feature space, they were eliminated for some category of the heatmaps. Table 2 shows the regional features set extracted for each category of heatmaps.

Regional and global features are combined to a features vector of 192 dimensions, and used for slide level training using XGBoost classifier.

**Table 2. Feature set extracted from each heatmap categories for training XGBoost slide level classifiers.**

| Features set extracted from heatmaps for slide level classifiers | | | | |
|---|---|---|---|---|
| Feature | Malignant heatmap | High Grade heatmap | Low Grade heatmap | Normal heatmap |
| area of the region | ✓ | ✓ | ✗ | ✗ |
| the bounding box area of the region | ✓ | ✓ | ✓ | ✓ |
| the major axis length of the ellipse with the same second moments of area as the region | ✓ | ✓ | ✓ | ✓ |
| the max intensity within the region | ✓ | ✓ | ✓ | ✗ |
| the mean intensity within the region | ✓ | ✓ | ✓ | ✓ |
| the min intensity within the region | ✓ | ✗ | ✓ | ✗ |
| area of convex hull | ✓ | ✓ | ✓ | ✗ |
| area of region with all holes filled | ✓ | ✓ | ✓ | ✓ |
| minor axis length of ellipse with same second moment of area | ✓ | ✓ | ✓ | ✓ |
| diameter of the region | ✓ | ✓ | ✗ | ✗ |
| Euler characteristic of the region | ✓ | ✓ | ✓ | ✗ |
| perimeter of the region | ✓ | ✓ | ✗ | ✗ |

**Training and predicting slide level classifier with XGBoost**. The extracted and refined features from the heatmaps are used to train the XGBoost as the slide level classifier. XGBoost is trained on the training set and then evaluated on the validation set. Since our datasets at slide level are also imbalanced, therefore we use Weighted XGBoost and pass weights for high grade, low grade and normal classes as 0.1 to the function. Other parameters of the XGBoost are defined as $n\_estimators$ = 60 and $max\_depth$ = 3. We do the training for 20 epochs to get the model with the highest accuracy and highest malignant sensitivity.

**Training and predicting slide level classifier with Random Forest**. Random Forest classifier is trained on the features extracted from the heatmaps for 300 epochs. These features were extracted from 3 largest clusters on malignant, high grade and low grade heatmaps generated for each slide. We use a grid search on the parameters (n_estimator, criterion, max_features, max_depth, min_samples_split, min_samples_leaf, bootstrap) of the classifier to find the best parameter set and ultimately use the best model with the best parameter set for prediction. Using Scikit-learn's *RandomizedSearchCV* method, we can define a grid of hyperparameter ranges, and randomly sample from the grid, performing K-Fold CV with each combination of values. Table 3 shows the parameters searched and the result of tuning the hyperparameters to find the best parameters for RF classifier for models trained on the features extracted from heatmaps generated at patch level with patch sizes 256 × 256 and 1024 × 1024 pixels.

**Table 3. Random search cross validation for hyperparameter tuning of Random Forest classifier.**

| Best parameters for Random Forest Classifier | | | |
|---|---|---|---|
| Parameter name | Parameter search range | Best parameters for Patch size (256 × 256) | Best parameters for Patch size (1024 × 1024) |
| n_estimators | [100, 110, 120, . . ., 1500] | 255 | 255 |
| min_samples_split | [2, 5, 10] | 10 | 2 |
| min_samples_leaf | [1, 2, 4] | 4 | 2 |
| max_features | ['auto', 'sqrt', 'log2'] | sqrt | sqrt |
| max_depth | [10, 20, 30, . . ., 110] | 10 | 30 |
| criterion | ['gini', 'entropy'] | gini | gini |
| bootstrap | [True, False] | True | True |

**Table 4. Distribution of samples in the test subset for inter-observer variation.**

| Category | SubCategory | Count | Total |
|---|---|---|---|
| Malignant | Squamous carcinoma | 25 | 50 |
| | Adenocarcinoma | 13 | |
| | CGIN | 7 | |
| | Other | 5 | |
| High Grade | CIN 2 | 27 | 50 |
| | CIN 3 | 23 | |
| Low Grade | HPV | 32 | 50 |
| | CIN 1 | 18 | |
| Normal/inflammation | Normal/inflammation | 50 | 50 |
| | Total Samples | | 200 |

## 3 Inter observer variations

In spite of the Bethesda system 2001 (TBS 2001) [21] formulating strict guidelines for reporting cervical smears, intra-observer and inter-observer variations are unavoidable and can be considered an inherent part of the reporting system [22].

The inter-observer variability in grading CIN [23] can add a layer of complexity to the training procedure using AI and cause an important issue for final diagnosis. CIN represents a morphological continuum, but biopsies displaying this lesion are classified into two or three grade categories [24]. The degree of agreement between two or more independent observers in the clinical setting constitutes interobserver reliability and is widely recognized as an important requirement for any behavioural observation procedure. Hence, a subset of test cervical biopsies (total 200 samples, 50 from each category balanced over subcategories) were re-annotated independently by three of the pathologists that annotated the whole slides for this project. Table 4 shows the distribution of inter-observer subset over category and sub-categories. Table 5 shows the agreement and disagreement of pathologists on the Categories and sub-categories on the 200 cervical samples. As this Table shows, the disagreement between pathologists is higher on sub-categories, which in some cases have lead to disagreement on categories.

The variations of the observations can be on different categories and sub-categories. Table 6 shows the result of comparing the original categories of these samples with the decision made by each of the observers. All three observers agree on all malignant cases, but there are some disagreements on other categories. The categories disagreements are sometimes because of disagreement in the sub-categories. Table 7, shows where the disagreement lies in subcategories.

### 3.1 Assessing reliability of annotations at slide level

In order to assess the reliability of the annotations, we use Cohen's kappa [25]. This statistic is used to measure the agreement between two observers. In the case of multiple observers, we can calculate Cohen's kappa for each pairs of observers [26] and then computed the arithmetic

**Table 5. Agreement and disagreement of the observers on the inter-observer subset.**

| Total Samples | Agreed on category | | Agreed on sub category | |
|---|---|---|---|---|
| | Yes | No | Yes | No |
| 200 | 177 | 23 | 168 | 32 |

**Table 6. Matching observers assigned categories with original categories.**

| Original Category | Observed Category | Observer A | Observer B | Observer C |
|---|---|---|---|---|
| Malignant | Malignant | 50 | 50 | 50 |
| | High Grade | 0 | 0 | 0 |
| | Low Grade | 0 | 0 | 0 |
| | Normal/inflammation | 0 | 0 | 0 |
| High Grade | Malignant | 0 | 0 | 0 |
| | High Grade | 49 | 48 | 47 |
| | Low Grade | 1 | 1 | 3 |
| | Normal/inflammation | 0 | 1 | 0 |
| Low Grade | Malignant | 0 | 0 | 0 |
| | High Grade | 3 | 1 | 2 |
| | Low Grade | 43 | 49 | 43 |
| | Normal/inflammation | 4 | 0 | 5 |
| Normal/inflammation | Malignant | 0 | 0 | 0 |
| | High Grade | 0 | 0 | 1 |
| | Low Grade | 1 | 6 | 3 |
| | Normal/inflammation | 49 | 44 | 46 |

**Table 7. Matching observers assigned sub categories with original sub categories.**

| | | Squamous carcinoma | Adeno carcinoma | CGIN | Other | CIN2 | CIN3 | CIN1 | HPV | Normal/inflammation |
|---|---|---|---|---|---|---|---|---|---|---|
| Observer A | Squamous carcinoma | 25 | - | - | - | - | - | - | - | - |
| | Adeno carcinoma | - | 12 | 1 | - | - | - | - | - | - |
| | CGIN | - | - | 7 | - | - | - | - | - | - |
| | Other | - | - | - | 5 | - | - | - | - | - |
| | CIN 2 | - | - | - | - | 25 | 1 | 1 | - | - |
| | CIN 3 | - | - | - | - | - | 23 | - | - | - |
| | CIN 1 | - | - | - | - | 2 | 1 | 14 | 1 | - |
| | HPV | - | - | - | - | - | - | - | 28 | 4 |
| | Normal/inflammation | - | - | - | - | - | - | 1 | - | 49 |
| Observer B | Squamous carcinoma | 25 | - | - | - | - | - | - | - | - |
| | Adeno carcinoma | - | 13 | - | - | - | - | - | - | - |
| | CGIN | - | - | 7 | - | - | - | - | - | - |
| | Other | - | - | - | 5 | - | - | - | - | - |
| | CIN 2 | - | - | - | - | 25 | - | 1 | - | 1 |
| | CIN 3 | - | - | - | - | - | 23 | - | - | - |
| | CIN 1 | - | - | - | - | 1 | - | 16 | 1 | - |
| | HPV | - | - | - | - | - | - | - | 32 | - |
| | Normal/inflammation | - | - | - | - | - | - | | 6 | 44 |
| Observer C | Squamous carcinoma | 25 | - | - | - | - | - | - | - | - |
| | Adeno carcinoma | - | 12 | 1 | - | - | - | - | - | - |
| | CGIN | - | - | 7 | - | - | - | - | - | - |
| | Other | 1 | 1 | 3 | - | - | - | - | - | - |
| | CIN 2 | - | - | - | - | 24 | - | 2 | 1 | - |
| | CIN 3 | - | - | - | - | - | 23 | - | - | - |
| | CIN 1 | | | | | 3 | - | 11 | 3 | 1 |
| | HPV | - | - | - | - | - | - | - | 28 | 4 |
| | Normal/inflammation | - | - | - | - | - | 1 | - | 3 | 46 |

mean of those values [27] for final Cohen's kappa over all the observations. In this experiment, we calculated Cohen's kappa over both categories and subcategories between the 3 observers. The Cohen's kappa score is 89.56% over categories and 87.24% over subcategories. The scores, measured for both cases, show a high level of agreement between the observers.

# 4 Results

In this section, we discuss the patch level and slide level results for cervical biopsies on training, validation and test sets. To show the effect of patch sizes on the classification results, the patches used for training are extracted from the WSIs with two different patch sizes, (256 × 256) and (1024 × 1024) pixels, all at highest magnification level (level 0 or 40X) with no overlap. We also have created two different balanced and imbalanced datasets to investigate the effect of including all patches in the training or excluding some patches from some categories on the training accuracy.

All the experiments are carried out on NVIDIA-DGX-1 with 8 Nvidia Tesla V100 GPUs. Training procedure is parallelised on 8 GPUs available to make efficient use of the resources and speed up the training.

## 4.1 Patch level results

The standard pretrained GoogLeNet model from torchvision library is used as patch level classifier.

Patches of size (256 × 256) and (1024 × 1024) pixels were extracted from tissue area of training and validation WSIs on a regular basis grid non-overlapping at highest magnification (40X) to create two independent patch datasets. The total number of extracted normal patches were more than the patches in other categories. Low grade and high grade extracted patches were less than normal and malignant patches, as shown in Table 8.

Table 8 shows the total number of patches extracted from the tissue regions of training and validation cervical WSIs for each category and the number of patches sampled for balanced and imbalanced sets from extracted patches for training the patch classifier.

To create the patch dataset for training, we sampled randomly from extracted patches of each category. We conducted experiments using two methods of sampling (balanced and imbalanced sampling). In balanced sampling, we sampled equal number of patches randomly from each category and used *"CrossEntropyLoss"* as our loss function while training. To have the balanced set of patches, some extracted patches were excluded for training. Since we want

**Table 8. Total extracted patches and used patched for training in each category for balanced and imbalanced patch dataset.** (patch sizes (256 × 256) and (1024 × 1024) pixels).

| Dataset | Category | Patch size (256 × 256) | | | Patch size (1024 × 1024) | |
|---|---|---|---|---|---|---|
| | | Total extracted patches | Sampled patches (Imbalanced) | Sampled patches (Balanced) | Total extracted patches | Sampled patches (Imbalanced) |
| Train | Malignant | 2,947,390 | 2,947,390 | 183993 | 225,415 | 225,415 |
| | High Grade | 210,109 | 210,109 | 183,993 | 23,397 | 23,397 |
| | Low Grade | 183,993 | 183,993 | 183,993 | 21,242 | 21,242 |
| | Normal | 4,529,685 | 2,947,390 | 1839,93 | 411,620 | 225,415 |
| | Total | 7,871,177 | 6,288,882 | 735,972 | 681,674 | 495,469 |
| Valid | Malignant | 1,484,692 | 1,484,692 | 87,026 | 115,580 | 115,580 |
| | High Grade | 102,929 | 102,929 | 87,026 | 11,046 | 11,046 |
| | Low Grade | 87,026 | 87,026 | 87,026 | 9,959 | 9,959 |
| | Normal | 2,169,250 | 1,484,692 | 87,026 | 205,008 | 115,580 |
| | Total | 3,843,897 | 3,159,339 | 348,104 | 341,593 | 252,165 |

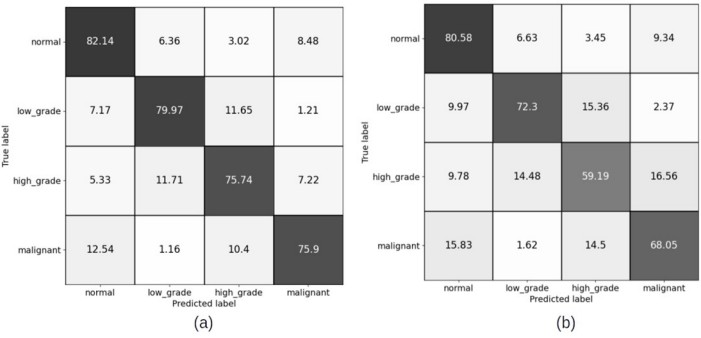

**Fig 7. Patch level confusion matrices for balanced cervical dataset.** (a) Train set confusion matrix
(Accuracy = 79.58%) (b) Validation set confusion matrix (Accuracy = 74.98%).

the performance tuned to high malignant sensitivity, it is preferred to use all the extracted
malignant patches in the training. Hence, in imbalanced sampling we used all the malignant
patches and sampled equal number of normal patches randomly and used all high grade and
low grade patches as they were less in number than malignant and normal patches. This
resulted in an imbalanced patch dataset. We used *"FocalLoss"* as our loss function while train-
ing to address the issue of the class imbalance problem for this method.

For the models trained on the patch datasets created using patches of size ($256 \times 256$) and
($1024 \times 1024$) pixels, in the following subsections the patch level confusion matrices for train-
ing and validation and test sets are displayed. The values in these confusion matrices are per-
centages of the patches from each category being classified correctly or misclassified as other
categories. In subsection 5.1, these confusion matrices are discussed and compared against
each other in details.

**4.1.1 Patch level confusion matrices for patch size ($256 \times 256$) pixels.** Figs 7 and 8 show
the results of training the patch classifier on the balanced and imbalanced datasets for patches
of size $256 \times 256$ pixels for training and validation sets, respectively on train and validation
sets.

**4.1.2 Patch level confusion matrices for patch size ($1024 \times 1024$ pixels) pixels.** Fig 9
shows the result of training the patch classifier on the imbalanced dataset for patches of size
$1024 \times 1024$ pixels for training and validation sets. For the case of patch size ($1024 \times 1024$), we
have not trained on balanced dataset. The reason is that the total number of extracted patches

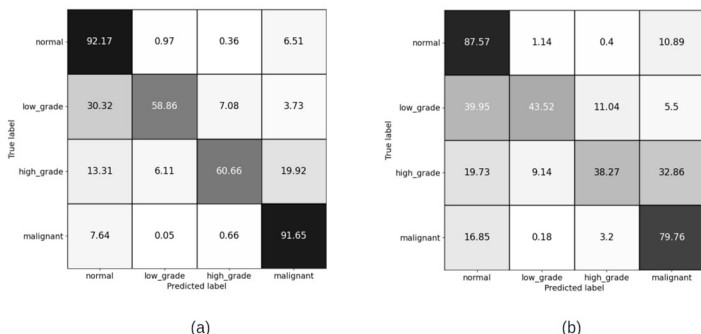

**Fig 8. Patch level confusion matrices for imbalanced cervical dataset.** (a) Train set confusion matrix
(Accuracy = 90.35%) (b) Validation set confusion matrix (Accuracy = 82.23%).

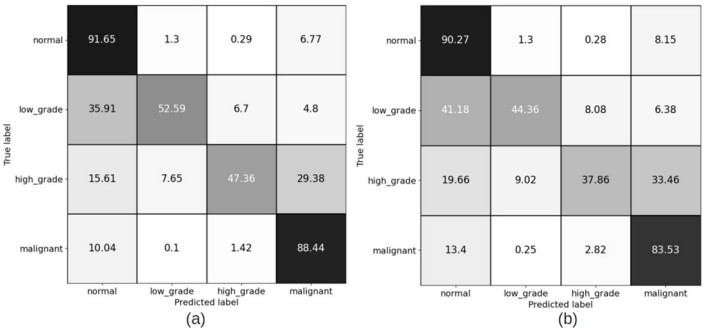

**Fig 9. Patch level confusion matrices for imbalanced cervical dataset.** (a) Training set confusion matrix (Accuracy = 87.85%) (b) Validation set confusion matrix (Accuracy = 84.96%).

for low grade and high grade categories is too few compared to normal and malignant. Hence, balancing the dataset means training on a very less number of patches, which is not ideal using deep neural networks.

**4.1.3 Patch level heatmaps.** The heatmaps shown in Figs 2, 3, 4, 5 and 6 are examples of correctly classified slides at slide level for different categories of cervical biopsies. In this section, we show the heatmaps at patch level for all the malignant slides in validation set that have been misclassified at slide level for imbalanced dataset with patch size 256 × 256 pixels. Each row in Fig 10 show the truth label, different category binary heatmaps and predicted heatmap for the malignant slides misclassified as high grade. Each row in Fig 11 show the truth label, different category heatmaps and predicted heatmap for the malignant slides misclassified as normal.

In Figs 2, 3, 4, 5, 6, 10 and 11, truth labels are the annotations displayed on the downsampled version of the image. Normal, low grade, high grade and malignant binary heatmaps are generated using the patch probabilities computed through inferencing the trained model on the tissue patches of each slide. Ultimately, the prediction heatmaps are, combining all 4 binary heatmaps into one thresholded by a value and shown in a colour per category.

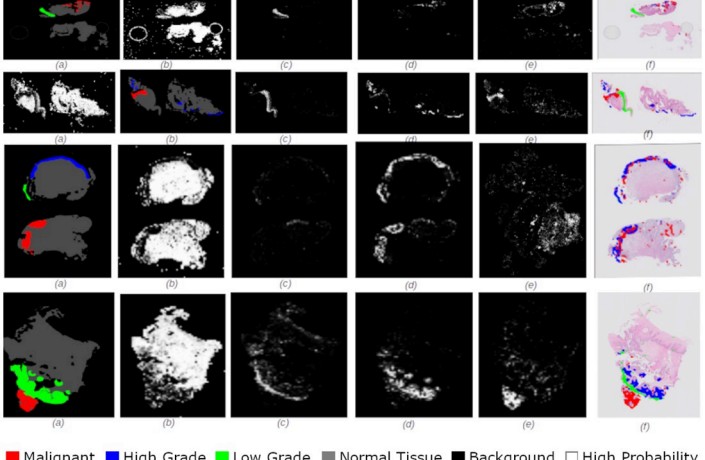

■ Malignant ■ High Grade ■ Low Grade ■ Normal Tissue ■ Background □ High Probability

**Fig 10. Patch level heatmaps for malignant slides classified as high grade slides.** (a):Truth Label (b): Normal (c): Low Grade (d): High Grade (e): Malignant (f): Prediction.

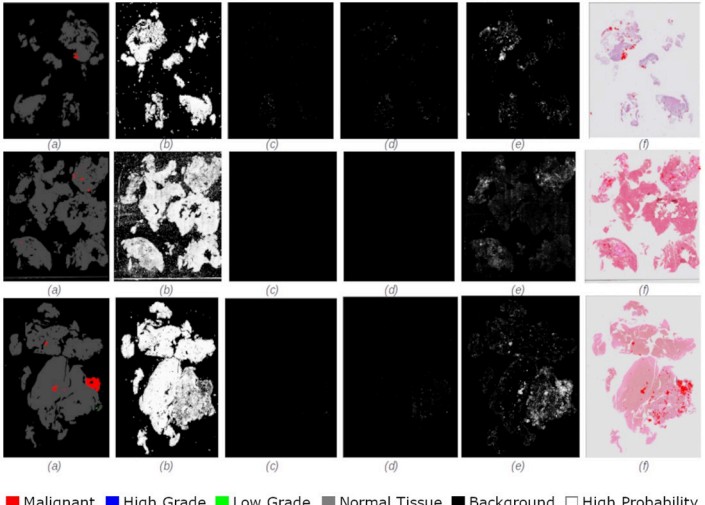

■ Malignant ■ High Grade ■ Low Grade ■ Normal Tissue ■ Background □ High Probability

**Fig 11. Patch level heatmaps for malignant slides classified as normal slides.** (a):Truth Label (b): Normal (c):Low Grade (d): High Grade (e): Malignant (f): Prediction.

## 4.2 Slide level results

To get the slide level diagnosis, patch level results needs to be aggregated using machine learning or CNN classifiers. We have extracted the features from the generated heatmaps at patch level and a trained two types of machine learning classifiers on the features to get the final diagnosis on the WSIs. XGBoost and Random Forest (RF) are the two classifiers that have been used in this experiment. Each of these classifiers are trained on the respective set of features extracted for these classifiers, as have been discussed in subsection 2.3.2.

**4.2.1 Slide level results for patch sizes (256 × 256) pixels.** The slide level results have been computed using XGBoost and Random Forest (RF) classifiers, for both balanced and imbalanced datasets. Figs 12 and 13 show the confusion matrices for balanced sets, and Figs 14 and 15 show the confusion matrices for imbalanced sets on training and validation sets for XGBoost and RF classifiers.

**4.2.2 Slide level results for patch sizes (1024 × 1024) pixels.** Figs 16 and 17 show the confusion matrices on train and validation imbalanced datasets using XGBoost and RF

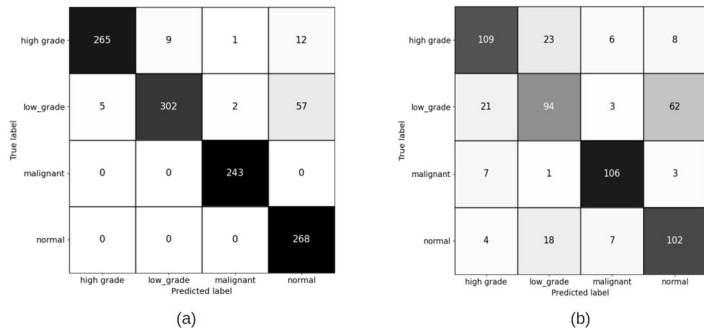

**Fig 12. Slide level confusion matrices for balanced cervical dataset using XGBoost classifier for patch size (256 × 256) pixels.** (a) Training set confusion matrix (Accuracy = 92.61%) (b) Validation set confusion matrix (Accuracy = 71.60%, Malignant sensitivity = 90.60%).

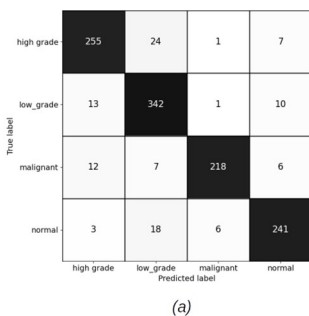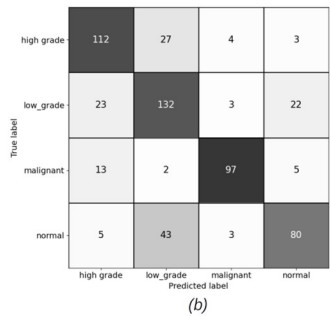

**Fig 13. Slide level confusion matrices for balanced cervical dataset using Random Forest classifier for patch size (256 × 256) pixels.** (a) Train set confusion matrix (Accuracy = 90.72%) (b) Validation set confusion matrix (Accuracy = 73.34%, Malignant sensitivity = 82.91%).

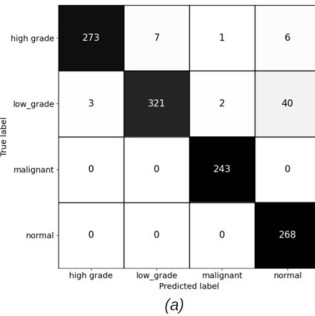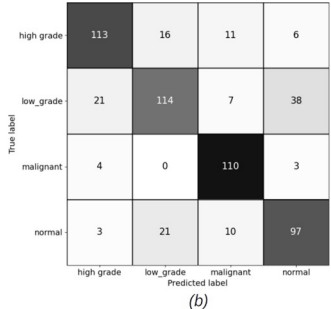

**Fig 14. Slide level confusion matrices for imbalanced cervical dataset using XGBooost classifier for patch size (256 × 256) pixels.** (a) Training set confusion matrix (Accuracy = 94.93%) (b) Validation set confusion matrix (Accuracy = 75.61%, Malignant sensitivity = 94.02%).

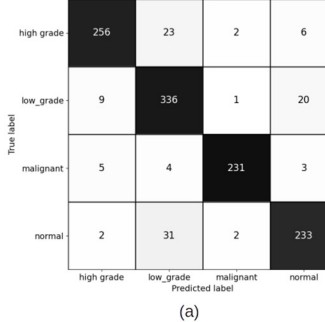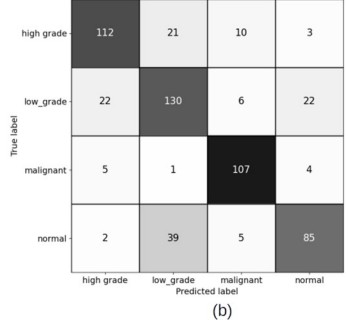

**Fig 15. Slide level confusion matrices for imbalanced cervical dataset using Random Forest classifier for patch size (256 × 256) pixels.** (a) Training set confusion matrix (Accuracy = 90.72%) (b) Validation set confusion matrix (Accuracy = 73.34%, Malignant sensitivity = 82.91%).

classifiers. We do not have a balanced set for patches of size (1024 × 1024) pixels. The reason for this has been discussed in earlier sections. The overall accuracy of the slide level classifier for both XGBoost and RF are same, but the malignant sensitivity at slide level for XGBoost is higher than RF due the larger number of the malignant slides classified correctly using XGBoost.

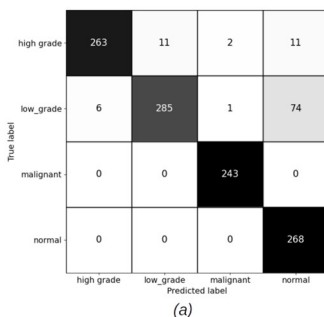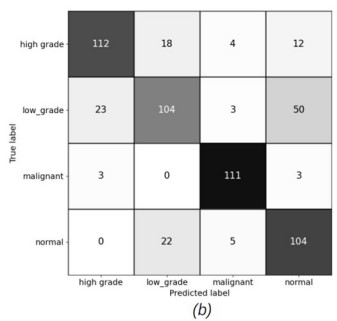

*(a)* *(b)*

**Fig 16. Slide level confusion matrices for imbalanced cervical dataset using XGBoost (Patch size = 1024 × 1024 pixels).** (a) Training set confusion matrix (Accuracy = 90.98%) (b) Validation set confusion matrix (Accuracy = 75.09%, Malignant sensitivity = 94.87%).

## 4.3 Test results

The trained models on the imbalanced dataset, with GoogLeNet classifier at patch level and XGBoost as slide classifier trained on a refined set of features extracted from heatmaps generated at patch level, were chosen as our best and final models. These rained models are then used to evaluate the performance of the trained model on the test set at patch level and slide level.

**4.3.1 Test patch level results.** Total number of patches in the training set for malignant and normal categories are equal for the imbalanced set and more than low grade and high grade patches. Hence, the patch classifier has been able to classify these categories better than the low grade and high grade categories and consequently more number of patches of these two categories are classified correctly in the test set.

The annotations at patch level are not very accurate, and therefore the malignant patches may contain some normal pixels and vice versa. This holds for patches from all categories. That is the reason for the patches from categories which have been wrongly classified as other categories. The overall accuracy of the patch level classifier over test set is close to the overall accuracy over validation set at patch level for the GoogLeNet model trained on patches of size (256 × 256) pixels. Both confusion matrices show that the patch classifier has been able to classify more number of normal and malignant patches correctly compared to the other categories. Fig 18 shows the confusion matrix over the test set at patch level for patch sizes (256 × 256).

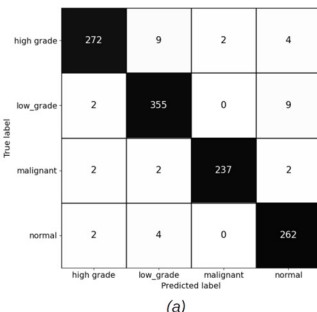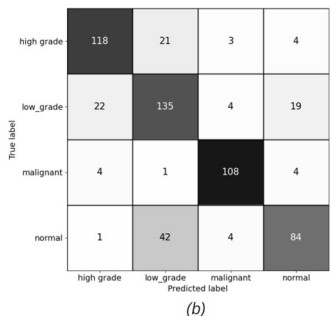

*(a)* *(b)*

**Fig 17. Slide level confusion matrices for imbalanced cervical dataset using Random Forest (Patch size = 1024 × 1024 pixels).** (a) Train set confusion matrix (Accuracy = 96.74%) (b) Validation set confusion matrix (Accuracy = 77.53%, Malignant sensitivity = 92.13%).

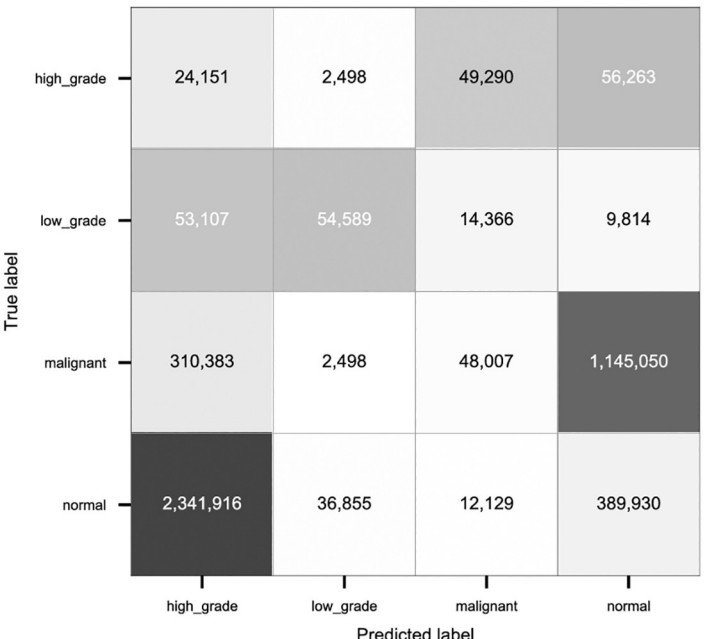

**Fig 18. Patch level confusion matrix for test set using GoogLeNet classifier (Patch size (256 × 256), Accuracy = 78.79%, Malignant sensitivity = 76.04%.**

**4.3.2 Test slide level results.** Fig 19 shows the confusion matrix over the test set for patch sizes 256 × 256 pixels. The trained patch level classifier was able to classify a high percentage of the malignant patches correctly in the test set (see Fig 18). The slide level classifier is also tuned toward getting higher malignant sensitivity. As the result, the malignant sensitivity over test set is high and more number of the malignant slides are classified correctly compared to the slides from other categories.

Figs 20 and 21 are examples of test cases correctly classified. Comparison of the prediction heatmaps with the ground truths in both figures show that the trained model has been able to classify the patches to a great extent and is able to classify the patches correctly for unseen data. The slide level classifier has been able to draw proper conclusions for slide level diagnosis of the test cases.

The pathologists that annotated the whole slide images believe, the potential time to examine the tissue and make a diagnosis on a case is approximately five minutes. The diagnosis time of our developed algorithm on the all cases in the test set, running on the NVIDIA Tesla V100 GPU, on average was 1.5 minutes per case. This algorithm runs in advance of a pathologist looking at the slide, and can save pathologist's time to a great extent.

**4.3.3 Comparison of our algorithm with state of the art algorithms.** Most of the available papers on cervical cancer detection are on CT/MRI/PET-CT, pop smear and cytology images and they perform different type of segmentation and classification tasks. A recent review of artificial intelligence in gynecological cancers [28] found thirty-four studies on cervical cancer, of which 18 used imaging data and 16 used value-based data. Three studies used MRI images to predict the stage of cervical cancer. Fifteen studies used CT/MRI/PET-CT images to predict recurrence and metastasis. Ten studies used clinical parameters to predict the diagnosis. Six studies used clinical parameters to predict therapeutic courses. The dataset used in our paper has not been used before and the task and the categories we perform

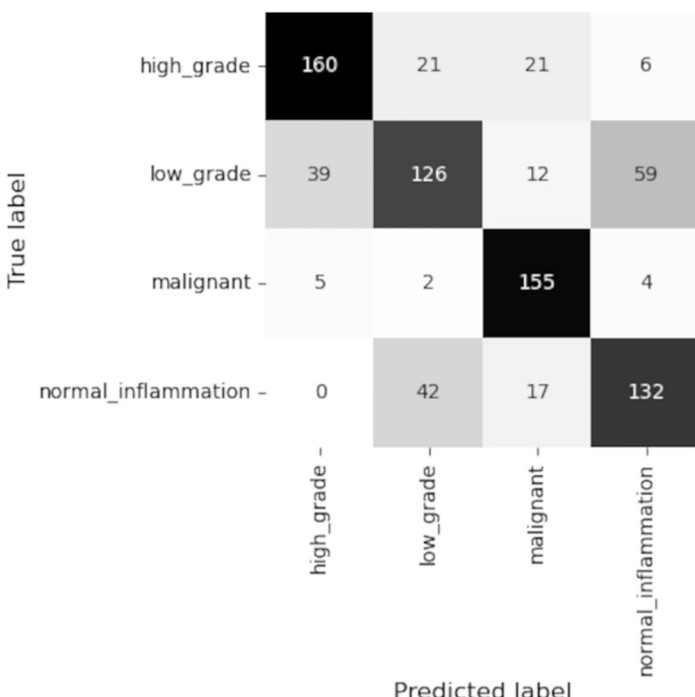

**Fig 19. Slide level confusion matrix for test set using XGBoost classifier(patch size 256 × 256), Accuracy = 71.53%, Malignant sensitivity = 93.40%.**

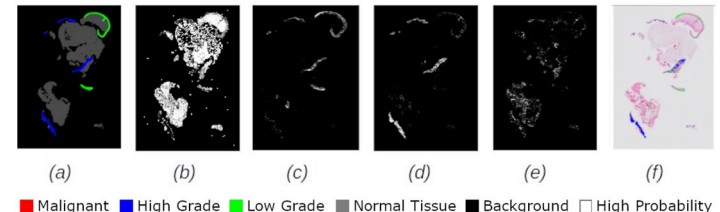

**Fig 20. Heatmaps for a test set high grade slide classified as high grade.** (a):Truth Label (b): Normal (c):Low Grade (d): High Grade (e): Malignant (f): Prediction.

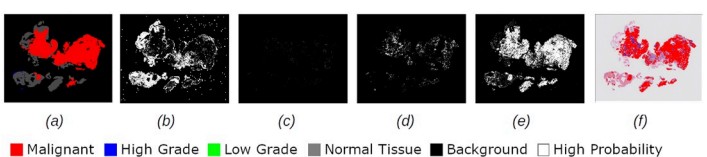

**Fig 21. Heatmaps for a test set malignant slide classified as malignant.** (a):Truth Label (b): Normal (c):Low Grade (d): High Grade (e): Malignant (f): Prediction.

**Table 9. Comparison of our algorithm with state of the art algorithms.**

| Paper | Dataset | Algorithm | Test Slides | Sensitivity |
|---|---|---|---|---|
| Our paper | Hematoxylin and Eosin Stained (H&E) Whole Slide Images | GoogLeNet + XGBoost | 801 | 93.40% |
| [11] | Cytology Whole Slide Images | RNN | 1170 | 95.1% |
| [29] | ThinPrep Cytologic | Mask R-CNN + XGBoost Test (TCT) images | 400 | 91.92% ± 5.2 96.21% ± 4.1 |

classification on, are different. Despite these facts, we try to compare the performance of our algorithm with some of the related state of the art papers.

The proposed algorithm in [11] is a progressive lesion cell recognition method combining low-and high-resolution WSIs to recommend lesion cells and a recurrent neural network-based (RNN) WSI classification model to evaluate the lesion degree of cervical WSIs. On independent test sets of 1,170 patient-wise WSIs, this proposed algorithm has reached 95.1% Sensitivity for classifying slides. In [29] a predictive model consists of two main phases is proposed. A Mask R-CNN model segments and classify cells in all ThinPrep Cytologic Test (TCT) images from patients into three classes and then a machine-learning model is used to classify patients into two classes (normal vs. abnormal), using the combination of cell classification from the best models constructed (T1, A3) and clinic diagnosis results. Table 9 shows comparison of the performance of our algorithm with some of the state of the art algorithms for cervical cancer classification considering the fact that the type of images used in these papers are different from ours.

## 5 Discussion

### 5.1 Patch level results discussion

Comparing the patch level confusion matrices for balanced and imbalanced datasets with patch size $256 \times 256$, in Figs 7 and 8, the trained model on larger dataset (imbalanced) has an overall higher accuracy for both training and validation sets. However, looking at the percentage of correctly classified patches in each category, the model trained on the balanced dataset has performed better on classifying the low grade and high grade categories. This is expected as the number of the patches in all categories are similar in the balanced case and therefore the model ins not biased to a specific category. But as we want to use all the malignant patches in the training procedure, and there is not enough patches in low grade and high grade categories compared to malignant and normal categories, we make use of FocalLoss to handle the imbalanced class training by adjusting the weights for hard or easily misclassified examples.

Although cell characteristics can be extracted from individual patches, but higher level structural information, such as the shape or extent of a tumour, can only be captured when analysing larger regions. Due to the limitation of the input size of the images that can be fed to CNNs, optimization challenges and graphical memory constraints, the whole WSI can not be fed to the CNN. To be able to make use of higher level structural information in the training, we decided to train the model with the largest patch size possible, extracted from WSIs. Hence, patches of 1024 pixels, the maximum size we could pass as input to the model considering the memory limitations, were extracted and used for training.

Comparison of the results of training the model on two different patch sizes, in Figs 8 and 9, show the improvement in overall accuracy over the validation set for larger patch sizes. The model trained on larger patch sizes is also able to better distinguish between the normal and malignant patches, while there is still a lot of confusion classifying neighbouring categories,

specifically low grade and high grade categories. The high grade and low grade categories are clinically so close and have a lot of histological overlap, and therefore confusion in distinguishing the patches of these categories was expected.

In summary, some reasons for the problems in accurately classifying the categories appropriately can be outlined as following:

- The histological overlapping of features at patch level makes it difficult for the AI algorithms to distinct different categories or sub-categories from each other, which results in misclassifying the patches and therefore affecting the final slide level prediction.

- In case of imbalanced datasets, the number of normal and malignant patches extracted and used for training have been far more than the low grade and high grade patches and therefore the trained model have learned the distinction features of these categories better.

- Each category contains sub-categories, and patches extracted for each category can be imbalanced between the subcategories as well. Imbalanced subcategories patches can affect the ability of the model to learn one subcategory better than the other with fewer patches in the dataset.

- The annotation precision also can affect the patch level results. Due to the fragmented nature of cervical biopsies, it is quite difficult to annotate them at pixel level precisely and most of the time annotations from one category may contain other subcategories or even categories in them. This leads to more complexity in distinction between the patches, specifically with smaller patch sizes.

Heatmaps generated at patch level, can support explainability of deep learning predictions in medical image analysis and provide clinicians with crucial visual cues that could ease their decision to accept or reject a deep learning based diagnosis. All the cases shown in subsection 2.3.2, are examples of the slides of different categories, that have been classified correctly. Comparing the truth label with prediction heatmaps of all these cases show, quite high percentage of patches have been classified correctly at patch level for slides containing only one sub-category in them (i.e. Figs 2, 3 and 4). This is an evidence of how well the patch classifier is able to distinguish between each of the categories and normal category.

Figs 5 and 6 are multi-label case containing more than one category in them. Comparing the truth label and prediction heatmaps for these cases show, how patch level classifier have been confused classifying some patches from neighbouring classes. These confusions are most likely due to histological overlap of neighbouring classes. The other reason is due to fewer number of patches in training set for some categories, the patch level has not been able to learn the distinction between features of the patches of neighbouring categories. Slide level classifier has been able to resolve these confusions and make the correct final slide level classification for these cases.

The slides, in Fig 10, are malignant slides that are misclassified as high grade. High-grade glandular abnormality (cervical glandular intraepithelial neoplasia (CGIN)), was put into the malignant category as it is usually treated more aggressively, but there can be histological overlap with some well differentiated adenocarcinomas that make it challenging to be differentiated. One possible solution was to consider CGIN and Adenocarcinoma as separate categories, but due to fewer cases of these two types, trying out this did not improve accuracy of our classification, and therefore we decided to keep the categories and sub-categories as were defined in Table 1.

The first and third rows in Fig 10 show heatmaps for malignant slides (CGIN) misclassified as high grade. CGIN is an in-situ/preinvasive lesion and at patch level and has some nuclear

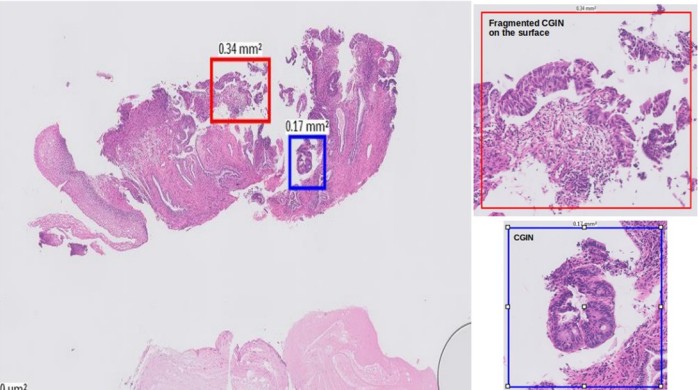

**Fig 22. Histological overlap between Fragmented CGIN, on the surface of the cervix and high grade CIN at patch level.**

features similar to the high grade areas. The red box in Fig 22 shows a fragmented CGIN area on the surface of the cervix. The features of it at patch level histologically overlaps with high grade CIN. The blue box shows a CGIN area which is not fragmented and correctly classified as malignant at patch level. There are malignant areas wrongly classified as high grade, as shown in Fig 10f in first and third rows, in blue colour. The histological overlapping of features at patch level as is mentioned in the beginning of this paragraph is the reason for this prediction and the probable reason for the slide level classifier to classify the slide wrongly at slide level.

The second case (second row) in Fig 10, is also a malignant case (squamous carcinoma) which is the most common type of cervical cancer. Comparing the truth label (Fig 10a in second row) with the prediction heatmap (Fig 10f in second row) show that the patch level classifier has performed well in picking up the high grade and malignant abnormalities. There is a small invasion in the malignant area, pointed by arrows in the red box in Fig 23. The low grade areas predicted (green areas in Fig 10f in second row), are just normal squamous with reactive changes that can be histological overlap with low grade lesions at a patch level.

The fourth case (fourth row) is a malignant (CGIN) slide. The predicted heatmap in Fig 10f in the fourth row, shows that malignant abnormality has been picked up correctly at patch

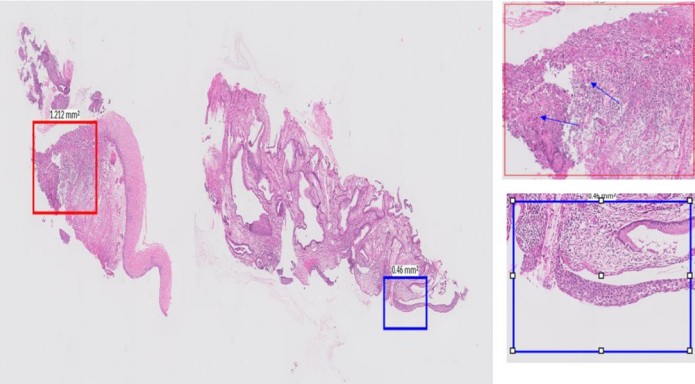

**Fig 23. Example of histological overlap between normal squamous with reactive changes and low grade lesions at patch level.**

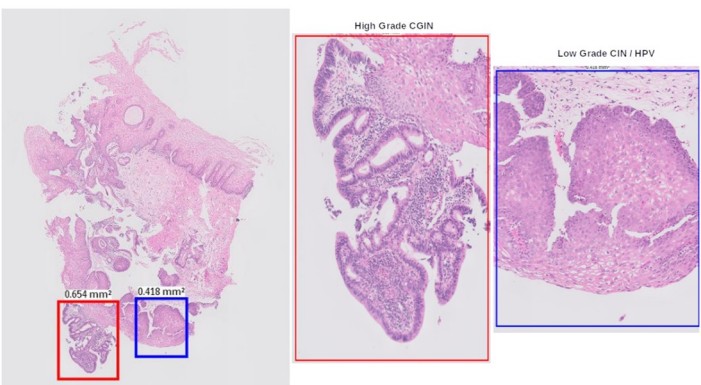

**Fig 24. Example of histological overlap between low grade and high grade at patch level.**

level (the area shown by the red box in Fig 24). The squamous area underneath the malignant area is low grade/viral. The area annotated as high grade in Fig 10a in the fourth row, contains some low grade CIN/HPV (the area shown by the blue box in Fig 24) which is predicted correctly in Fig 10f in the fourth row. The other parts of high grade wrongly classified as low grade can be the result of histological overlap between low grade and high grade at patch level.

Fig 11 shows examples of malignant slides are misclassified as normal. Most of these slides contain small fragments of tumour which are picked up correctly by the patch level classifier (small tumour fragments within blood and fibrinoid material, red boxes in Fig 25). There are other areas picked up by the classifier as malignant tissue that are non-diagnostic in isolation (blue box with necrosis in Fig 25). The proportion of malignant patches on these slides is much smaller than the proportion of normal patches. Even though the patch classifier has been able to classify the malignant patches on these slides, the slide level classifier has not been able to draw the final correct decision for these slides.

## 5.2 Discussion on slide level results

For $256 \times 256$ patches, the slide level malignant sensitivity using XGBoost classifier as slide level classifier is 90.6% and using RF is 82.91% on validation set for balanced cervical dataset and for imbalanced datasets is 94.02% for XGBoost and 91.45% for RF.

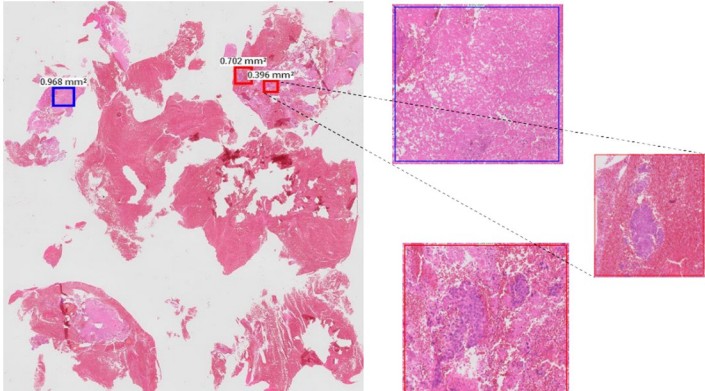

**Fig 25. Small tumour fragments with blood and fibrinoid materials.**

For 1024 × 1024 patches, the malignant sensitivity for XGBoost classifier at slide level is 94.87% and for RF is 92.31% on validation set for imbalanced cervical dataset.

Although overall accuracy of RF at slide level is higher than XGBoost for both patch sizes on the balanced dataset, RF has misclassified some more of malignant slides for normal, high grade and low grade categories. For the imbalanced dataset, overall accuracy of both the slide level classifiers are the same, but XGBoost has been able to classify a higher number of the malignant slides correctly. Hence, we can conclude that XGBoost classifier has an overall better performance than RF for being able to recognise more numbers of malignant slides correctly.

## 6 Conclusion

In this paper, we present an AI algorithm, trained and evaluated on cervical biopsies. The aim is to increase overall efficiency of pathological diagnosis by having the algorithm detect common patterns. We also want to have the performance tuned to high sensitivity for malignant cases, which could be expedited for pathologist assessment. The dataset being imbalanced, annotation limitations and the difficulty in distinction between some categories and sub-categories due to similarity in morphological structures, add layers of complexity to the training procedure. Despite all these, the developed algorithm performed well in classifying malignant cases against the other categories, and it reached 93.40% malignant sensitivity for classifying slides on test dataset. The performance of the algorithm on the low grade and high grade categories can be improved by introducing more variations of these cases to the training set. There is potential for the developed algorithm to be a useful clinical tool for pathologists and warrants further validation at larger scale. The code and the trained models for this paper is available at [30].

## Acknowledgments

We acknowledge the support of NHS Research Scotland (NRS) Greater Glasgow and Clyde Biorepository. We acknowledge the support of the biomedical scientists, Tim Prosser, Lucy Irving, Jennifer Campbell and Jennifer Faulkner, from the Pathology Department, NHS Greater Glasgow and Clyde for their work in identifying blocks, the technical work generating and scanning slides, as well as annotating slides. Additionally, William Sloan of NHS Greater Glasgow and Clyde Biorepository without whose work in tracking the scanned images and annotations this study could not have been completed in a timely manner.

### Ethical approval

This study uses archived samples from the NHS Greater Glasgow and Clyde Biorepository. Patients gave informed consent for surplus tissue to be stored and used for medical research, this consent was recorded in an electronic Surplus Tissue Authorisation form. All data was de-identified within the biorepository and as provided to the researchers was fully anonymised.

- Ethics approval for the study was granted by NHS Greater Glasgow and Clyde Biorepository and Pathology Tissue Resource (REC reference 16/WS/0207) on 4th April 2019.

- Biorepository approval was obtained (application number 511)

- Local approval was obtained from the School of Computer Science Ethics Committee, acting on behalf of the University Teaching and Research Ethics Committee (UTREC) [Approval code- CS15840].

## Author Contributions

**Conceptualization:** Mahnaz Mohammadi, Gareth Bryson.

**Data curation:** Mahnaz Mohammadi, Sheeba Syed, Prakash Konanahalli, Sarah Bell, Gareth Bryson.

**Formal analysis:** Mahnaz Mohammadi, Christina Fell.

**Funding acquisition:** David J. Harrison.

**Investigation:** Mahnaz Mohammadi, Christina Fell.

**Methodology:** Mahnaz Mohammadi, Christina Fell.

**Project administration:** Sarah Bell.

**Resources:** Gareth Bryson.

**Software:** Mahnaz Mohammadi, Christina Fell, David Morrison.

**Supervision:** David Harris-Birtill.

**Validation:** Mahnaz Mohammadi.

**Visualization:** Mahnaz Mohammadi.

**Writing – original draft:** Mahnaz Mohammadi.

**Writing – review & editing:** Mahnaz Mohammadi, Christina Fell, Sarah Bell, Gareth Bryson, Ognjen Arandjelović, David J. Harrison, David Harris-Birtill.

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
