## [Decision Letter · Decision Letter 0]

9 Jun 2023

PDIG-D-22-00313

Automated reporting of cervical biopsies using artificial intelligence

PLOS Digital Health

Dear Dr. Mohammadi,

Thank you for submitting your manuscript to PLOS Digital Health. We apologize for the severe delay during the review process; unfortunately, a few agreed reviews were ultimately not submitted, and it was necessary to make an editorial assessment based on the one review obtained. After careful consideration, the editors feel that the manuscript has merit but does not fully meet PLOS Digital Health's publication criteria as it currently stands. Therefore, we invite you to submit a revised version of the manuscript that addresses the points raised during the review process.

Please submit your revised manuscript within 30 days Jul 09 2023 11:59PM. If you will need more time than this to complete your revisions, please reply to this message or contact the journal office at digitalhealth@plos.org. Please include the following items when submitting your revised manuscript:

We look forward to receiving your revised manuscript.

Kind regards,

Adeline Boatin

Guest Editor

PLOS Digital Health

Journal Requirements:

a. State what role the funders took in the study. If the funders had no role in your study, please state: “The funders had no role in study design, data collection and analysis, decision to publish, or preparation of the manuscript.”

b. If any authors received a salary from any of your funders, please state which authors and which funders.

3. We ask that a manuscript source file is provided at Revision. Please upload your manuscript file as a .doc, .docx, .rtf or .tex.

4. Please provide separate figure files in .tif or .eps format only and remove any figures embedded in your manuscript file. Please also ensure that all files are under our size limit of 10MB.

Additional Editor Comments (if provided):

Reviewers' comments:

Reviewer's Responses to Questions

**Comments to the Author**

1. Does this manuscript meet PLOS Digital Health’s publication criteria? Is the manuscript technically sound, and do the data support the conclusions? The manuscript must describe methodologically and ethically rigorous research with conclusions that are appropriately drawn based on the data presented.

Reviewer #1: Yes

2. Has the statistical analysis been performed appropriately and rigorously?

Reviewer #1: Yes

3. Have the authors made all data underlying the findings in their manuscript fully available (please refer to the Data Availability Statement at the start of the manuscript PDF file)?

Reviewer #1: No

4. Is the manuscript presented in an intelligible fashion and written in standard English?

Reviewer #1: Yes

5. Review Comments to the Author

Reviewer #1: The paper presents development of an Artificial Intelligence (AI) algorithm for automated diagnosis of cervical cancer from cervical biopsies. The paper presents an interesting topic given the high prevalence of cervical cancer globally; and the need for early and accurate diagnosis. The paper acknowledges that previous work has been done in the field, but mentions some of their limitations including manually extracting features (handcrafted), that need expert domain knowledge and the procedure is laborious and time-consuming. The authors have used a number of recent articles; and the data acquisition, curation, feature selection and classification steps are very clear.

However, the following comments should be addressed to make the manuscript more readable.

Please indicate out of the 2539 whole slide images (WSIs), the number of images that were obtained from Glasgow Royal Infirmary (NG), Southern General Hospital (SG), Royal Alexandria Hospital (RAH) and Queen Elizabeth University Hospital (QEUH).

Please indicate the imaging parameters of the Philips UFS Scanner including resolution, FOV; and the Magnification of the equipment.

On line 140, you mention that two of the labs (labs 6 & 8) and 10% of WSIs from other labs which were randomly. Please indicate which are the labs 6 & 8 mentioned in the manuscript.

Justify why you selected 33% of the training slides as your test train split. Was increasing or reducing this affecting your results in any way (Line 142). However, your Table 1 indicates otherwise. Please clearly explain your test train split.

Avoid putting tables in between a sentence. Place Table 2 after paragraph 297 to improve readability.

XGBoost classifier is an excellent algorithm that has been used by several other authors. Can you please compare your results (probably some metrics of the confusion matrix) with some of the state of art available algorithms with a special focus on the XGBoost classifier variations proposed by different authors.

Are the datasets available for public use? This would really be very helpful to other researchers who may wish to validate their algorithms on this dataset.

Include the computational requirements for your algorithms; and the anticipated time to save while using the algorithms vs the manual process.

6. PLOS authors have the option to publish the peer review history of their article (what does this mean?). If published, this will include your full peer review and any attached files.

**Do you want your identity to be public for this peer review?** For information about this choice, including consent withdrawal, please see our Privacy Policy.

Reviewer #1: No

---

## [Decision Letter · Decision Letter 1]

3 Oct 2023

Automated reporting of cervical biopsies using artificial intelligence

PDIG-D-22-00313R1

Dear Dr Mohammadi,

We are pleased to inform you that your manuscript 'Automated reporting of cervical biopsies using artificial intelligence' has been provisionally accepted for publication in PLOS Digital Health.

Best regards,

Adeline Boatin

Guest Editor

PLOS Digital Health

No comments

Reviewer Comments (if any, and for reference):

Reviewer's Responses to Questions

**Comments to the Author**

1. If the authors have adequately addressed your comments raised in a previous round of review and you feel that this manuscript is now acceptable for publication, you may indicate that here to bypass the “Comments to the Author” section, enter your conflict of interest statement in the “Confidential to Editor” section, and submit your "Accept" recommendation.

Reviewer #1: All comments have been addressed

2. Does this manuscript meet PLOS Digital Health’s publication criteria? Is the manuscript technically sound, and do the data support the conclusions? The manuscript must describe methodologically and ethically rigorous research with conclusions that are appropriately drawn based on the data presented.

Reviewer #1: Yes

3. Has the statistical analysis been performed appropriately and rigorously?

Reviewer #1: Yes

4. Have the authors made all data underlying the findings in their manuscript fully available (please refer to the Data Availability Statement at the start of the manuscript PDF file)?

Reviewer #1: Yes

5. Is the manuscript presented in an intelligible fashion and written in standard English?

Reviewer #1: No

6. Review Comments to the Author

Reviewer #1: The paper has really improved and is now more readable and clearer.

7. PLOS authors have the option to publish the peer review history of their article (what does this mean?). If published, this will include your full peer review and any attached files.

**Do you want your identity to be public for this peer review?** For information about this choice, including consent withdrawal, please see our Privacy Policy.

Reviewer #1: **Yes: **Dr. Wasswa William (PhD)
